# Advances in the Formulation and Assembly of Non-Cationic Lipid Nanoparticles for the Medical Application of Gene Therapeutics

**DOI:** 10.3390/nano11030825

**Published:** 2021-03-23

**Authors:** Richard K. Fisher, Phillip C. West, Samuel I. Mattern-Schain, Michael D. Best, Stacy S. Kirkpatrick, Raymond A. Dieter, Joshua D. Arnold, Michael R. Buckley, Michael M. McNally, Michael B. Freeman, Oscar H. Grandas, Deidra J. H. Mountain

**Affiliations:** 1Department of Surgery, University of Tennessee Graduate School of Medicine, 1924 Alcoa Highway Box U-11, Knoxville, TN 37920, USA; RFisher@utmck.edu (R.K.F.III); PWest@utmck.edu (P.C.W.); skirkpat@utmck.edu (S.S.K.); rdieter@utmck.edu (R.A.D.III); jarnold@utmck.edu (J.D.A.); mbuckley@utmck.edu (M.R.B.); mmcnally@utmck.edu (M.M.M.); mfreeman@utmck.edu (M.B.F.); ograndas@utmck.edu (O.H.G.); 2Department of Chemistry, University of Tennessee Knoxville, 1420 Circle Drive, Knoxville, TN 37996, USA; SMattern@utmck.edu (S.I.M.-S.); best@ion.chem.utk.edu (M.D.B.)

**Keywords:** gene therapy, non-viral gene delivery, lipid nanoparticles, non-cationic liposomes, siRNA encapsulation, cell penetrating peptides, cellular uptake

## Abstract

Lipid nanoparticles have become increasingly popular delivery platforms in the field of gene therapy, but bench-to-bedside success has been limited. Many liposomal gene vectors are comprised of synthetic cationic lipids, which are associated with lipid-induced cytotoxicity and immunogenicity. Natural, non-cationic PEGylated liposomes (PLPs) demonstrate favorable biocompatibility profiles but are not considered viable gene delivery vehicles due to inefficient nucleic acid loading and reduced cellular uptake. PLPs can be modified with cell-penetrating peptides (CPPs) to enhance the intracellular delivery of liposomal cargo but encapsulate leakage upon CPP-PLP assembly is problematic. Here, we aimed to identify parameters that overcome these performance barriers by incorporating nucleic acid condensers during CPP-PLP assembly and screening variable ethanol injection parameters for optimization. CPP-PLPs were formed with R8-amphiphiles via pre-insertion, post-insertion and post-conjugation techniques and liposomes were characterized for size, surface charge, homogeneity, siRNA encapsulation efficiency and retention and cell associative properties. Herein we demonstrate that pre-insertion of stearylated R8 into PLPs is an efficient method to produce non-cationic CPP-PLPs and we provide additional assembly parameter specifications for a modified ethanol injection technique that is optimized for siRNA encapsulation/retention and enhanced cell association. This assembly technique could provide improved clinical translation of liposomal based gene therapy applications.

## 1. Introduction

Ribonucleic acid interference (RNAi) is a biological process that inhibits the post-transcriptional expression of mRNA in the cytoplasm of eukaryotic cells. The gene-specific nature of small-interfering RNA (siRNA), an active biological molecule of RNAi, provides a therapeutic avenue for silencing abnormal or unwanted gene expression often exemplified in vascular disease, cancers and other rare genetic disorders. However, siRNA is vulnerable to enzymatic degradation in vivo and lacks the ability to traverse cell membranes due to large molecular weight and hydrophilic properties. Therefore, the issue of delivery remains a primary hurdle to clinical application of siRNA therapeutics.

Liposomes are the most widely studied nonviral gene vectors, with the ability to encapsulate nucleic acid cargo within an aqueous compartment, protecting it from the external environment [1]. Many multifunctional modification strategies aimed at improving the clinical efficacy of liposome-mediated gene therapy have been elucidated [2]. This includes the incorporation of polyethylene glycol (PEG) on the surface, resulting in PEGylated liposomes (PLPs) with increased stability and half-life of siRNA cargo in vivo (2nd generation liposomes) [3,4] and the development of ligand-conjugated liposomes for enhanced cell uptake and targeted drug delivery (3rd generation liposomes) [5,6]. The use of cationic lipids in liposomal formulations (CLPs) has helped improve encapsulation of negatively charged nucleic acid cargo through electrostatic-mediated “lipoplexes” [7]. However, many synthetic, “unnatural” cationic lipids are associated with cytotoxicity through toll-like receptor activation [8], mitochondrial membrane disruption [9] and induction of reactive oxygen species [10]. Due to the cytotoxic effects of cationic liposomes, along with rapid opsonization and macrophage clearance, translational success of CLP-mediated gene therapy has been limited despite approval of numerous Phase I clinical trials [11]. An alternative strategy would be constructing nucleic acid lipoplexes with “natural” noncationic lipids in the presence of divalent cations (i.e., calcium; Ca^2+^), whereby electrostatic bridging between the phosphate backbone of nucleic acid and the zwitterionic phospholipid head group enhances nucleic acid loading during self-assembly [12]. However, inadequate cell uptake of standard PLP assemblies has limited basic science investigations using noncationic lipid formulations [13]. Cell-penetrating peptides (CPPs) have been incorporated into PLP bilayers to promote membrane translocation of liposomal cargo and subsequent cellular uptake [14,15]. Unfortunately, due to their inherent membrane agitation, the incorporation of CPP-amphiphiles into lipid bilayers of PLP formulations often results in cargo leakage and heterogeneous size dispersity [16]. Consequently, most of the techniques used to assemble CPP-modified PLPs with siRNA or other nucleic acid cargo are inefficient and costly.

Using octaarginine (R8), a well-established CPP [17], we aimed to elucidate optimal process parameters required to assemble noncationic, CPP-modified PLPs with efficient siRNA loading and functionally enhanced transfection capacity. Modification strategies for incorporating R8 into non-cationic PLPs (R8-PLP) were surveyed to empirically derive an assembly method that maximizes siRNA retention and encapsulation efficiency (EE%), while producing liposomal nanoparticles with critical quality attributes necessary for in vivo translation (Table 1). This strategy could provide the ground-breaking advancement required to augment bench-to-bedside success of liposomal-based RNAi technology. 

## 2. Materials and Methods

### 2.1. PLP Assembly via Ethanol Injection Technique

#### 2.1.1. Liposome Constituents

All liposome formulation constituents are defined in Table 2. Lipids and cholesterol were purchased from Avanti Polar Lipids (Alabaster, AL, USA). Custom R8 peptide was purchased with an azido-modified lysine (RRRRRRRRK(N3)-NH2) from P3 Biosystems (Louisville, KY, USA). Stearylated octaarginine (STR-R8) was purchased from LifeTein LLC (Somerset, NJ, USA). siRNA specific for constitutively expressed glyceraldehyde 3-phosphate dehydrogenase (GAPDH) was used as our universal nucleic acid residue for the derivation and optimization of standard assembly parameters (ThermoFisher Scientific, Waltham, MA, USA).

#### 2.1.2. Liposome Assembly

Base PLP nanoparticles were formed with bulk lipid DOPC:chol at 7:3 mol plus 10 mol% DSPE-PEG and were assembled using a previously described EtOH injection technique [18]. Briefly, lipids were dissolved in CHCl_3_, combined as indicated and dried under N_2_ gas and vacuum to remove remaining solvent. Dried lipids were then resuspended in molecular grade 100% EtOH. GAPDH siRNA at 20–50 ug/300 uL 10 mM Tris-HCl, pH 8.0 with 0–50 mM CaCl_2_ was injected with 200–5000 ug total lipid/200 uL 100% EtOH, under constant vortexing at room temperature (RT). Liposomes were purified from EtOH and un-encapsulated siRNA via 24 h dialysis against PBS, pH 7.4 at 4 °C. Liposomes were extruded using 100 nm polycarbonate NanoSizer^TM^ extruders from T&T Scientific prior to characterization (Knoxville, TN, USA).

### 2.2. CPP Modification via R8 Amphiphile Incorporation

#### 2.2.1. R8-PEG synthesis

DSPE-PEGdbco, a form of DSPE-PEG with a cyclooctyne modification commonly used in azide-alkyne cycloaddition reactions (i.e., click chemistry), was used to form R8-PEG amphiphiles for PLP incorporation via pre- and post-insertion. Briefly, equimolar azido-modified R8 peptides and DSPE-PEGdbco were combined at RT under constant agitation for 2 h, according to previously established reaction conditions [19,20,21]. MALDI-TOF Mass spectrometry confirmed peptide conjugation and reaction efficiency, whereby peaks of aggregate molecular weight were present in reaction product spectra compared to individual reactant spectra alone (Figure 1).

#### 2.2.2. PLP Modification with R8-PEG or STR-R8

For all modification strategies, base PLPs were assembled via EtOH injection as described and were modified by substituting DSPE-PEG or 7:3 DOPC/chol with R8-PEG or STR-R8 amphiphiles at equal mol%, respectively. In this way mol% PEG was kept constant across all conditions to control PEG-mediated membrane stability and assay encapsulate retention as a function of R8 modification alone. 

##### Modification via Pre-Insertion

R8-PEG or STR-R8 were combined at 1–10 mol% with base PLP lipid constituents at the time of lipid drying under N_2_ gas. Likewise, lipid hydration with EtOH was performed in one-step and all lipid constituents were incorporated at the time of initial liposome assembly. Liposomes were purified from un-encapsulated siRNA as described, prior to extrusion and characterization.

##### Modification via Post-Insertion 

Base PLPs were assembled as described without the incorporation of R8 amphiphiles. Following PLP purification by dialysis, R8-PEG or STR-R8 were combined with pre-formed base PLPs and incubated at 4 °C overnight or 37 °C for 4 h, according to previously established conditions for lipid transfer. A second overnight dialysis was performed following R8-amphiphile insertion to remove any un-retained siRNA encapsulate, prior to extrusion and characterization. 

##### Modification via Post-Conjugation 

Base PLPs were assembled as described, without the incorporation of R8, but with DSPE-PEGdbco added at the time of lipid drying. Following, purification by dialysis, azido-modified R8 peptides were added in an equimolar amount to DSPE-PEGdbco and incubated at 4 °C overnight or 37 °C for 4 h, according to previously established parameters of the azide-alkyne cycloaddition reaction. A second overnight dialysis was performed following R8 conjugation to remove any un-retained siRNA encapsulate, prior to extrusion and characterization.

### 2.3. Liposome Characterization Studies

#### 2.3.1. Size, Homogeneity and Charge Characterization

The mean size, associated polydispersity index (PDI) and zeta potential of all liposome preparations were measured following dialysis in PBS by dynamic light scattering and relative electrophoretic mobility in water using the Zetasizer Nano ZS instrument (Malvern Instruments Ltd., Worchestershire, UK).

#### 2.3.2. Quantitative siRNA Encapsulation Efficiency (EE%)

The siRNA encapsulation and retention of all liposome preparations was determined using the Quant-iT RiboGreen RNA Assay Kit (Thermofisher Scientific). Following dialysis purification from un-encapsulated and/or un-retained siRNA, liposomes were denatured and solubilized in 1% Triton X-100 at 37 °C for 15 min to release siRNA cargo. Released siRNA was mixed with RiboGreen reagent for fluorescent labeling and emission was read at 525 nm. Fluorescence units of solubilized liposomes was fit to a known standard curve of siRNA in 1% Triton X-100. EE% of each liposome formulation was calculated as (pmols siRNA encapsulate/total original pmols siRNA used) × 100.

#### 2.3.3. Gel Electrophoresis

The stability of siRNA-loaded liposomes was assessed using multiple gel electrophoresis techniques. Stability assays were run using 1% agarose gels with 1 ug/mL Ethidium Bromide (EtBr) and 1× Tris-Boric-EDTA (TBE) buffer for gel preparation and running. All siRNA bands were visualized by EtBr binding via transillumination through a GFP filter.

##### Gel Exclusion Assay

For a qualitative analysis of EE%, siRNA-loaded liposome sample aliquots were taken prior to purification by dialysis and equal fractions were loaded onto agarose gels. Un-encapsulated, un-complexed siRNA, free in solution, was ran into gel at 100 V for 30 min while liposomes with encapsulated/complexed siRNA were excluded from gel penetration and isolated to the wells. 

##### Heparin Displacement Assay

To determine minimum heparin concentration necessary for complete siRNA displacement from liposome surface, siRNA-loaded liposomes from 100:1 lipid:siRNA wt-to-wt samples were exposed to varying concentrations of heparin (0–400 ug/mL) for 30 min at room temperature. Equal fractions were loaded onto agarose gels and displaced siRNA was ran into gel at 100 V for 30 min. In follow-on experiments, to examine the proportion of lipoplexed siRNA that was outer associated and vulnerable to heparin displacement, liposome samples were taken after purification by dialysis, exposed to 100 ug/mL heparin and equal fractions were loaded and ran as described. To examine total encapsulated siRNA and total lipoplexed siRNA, liposome samples were taken after purification by dialysis, exposed to 1% Triton X-100 and 100 ug/mL heparin to denature liposome nanoparticles and release all complexed siRNA and equal fractions were loaded and ran as described. To examine the proportion of encapsulated siRNA that was completely contained within the core of the liposome, liposome samples were taken after purification by dialysis, treated with 100 ug/mL heparin and re-dialyzed overnight to remove previously complexed siRNA from solution, then exposed to 1% Triton X-100 and 100 ug/mL heparin to denature liposome nanoparticles and release all siRNA within the core. 

##### RNase Stability Assay

To determine minimum RNase necessary for complete siRNA degradation, free siRNA controls were exposed to varying concentrations of RNase A enzyme (0.001–100 ug/mL) for 30 min at 37 °C. Equal fractions were loaded onto agarose gels and the remaining, undigested siRNA was ran into gel at 100 V for 30 min. In follow-on experiments, siRNA-loaded liposomes were prepared with and without the exposure to 0.5 ug/mL RNase A, RNase digestion was stopped by flash freezing in liquid N_2_ and liposome samples were exposed to 1% Triton X-100 and 100 ug/mL heparin to denature liposome nanoparticles and release all complexed siRNA. The protection of outer associated siRNA demonstrated via heparin displacement was validated by the protection from RNase enzyme degradation compared to the degradation of free siRNA controls.

#### 2.3.4. Morphological Analysis by Scanning Transmission Electron Microscopy (STEM)

Liposome morphology and lamellarity were investigated by STEM using a negative-stain method. Liposomes were applied dropwise to a carbon film coated copper grid and allowed to air dry. Liposome films were then stained with 2% phosphotungstic acid and air-dried for 1 min at room temp. Samples were visualized with Zeiss Auriga 40 STEM scope and images were acquired by SmartSEM image acquisition software (Carl Zeiss, Inc., Oberkochen, Germany).

### 2.4. Vascular Smooth Muscle Cell Culture

Human aortic smooth muscle cells (HASMCs) were obtained from LifeLine Cell Technology (Walkersville, MD) as cryopreserved primary cultures of 49 years old male single-donor cells. Cells were incubated at 37 °C in an environment of 5% CO_2_ and 95% humidity and grown in VascuLife growth medium (VascuLife Basal Medium + VascuLife smooth muscle cell supplement kit + gentamycin/amphotericin; LifeLine Cell Technology). Prior to experimental use, a quiescent state was induced using Dulbecco’s Modified Eagle Medium (DMEM; Thermofisher Scientific) + gentamycin/amphotericin overnight.

### 2.5. Cell Association Studies

To measure cell association, liposomes were assembled as described with the addition of Rho-PE at 0.5 mol%. At ~80% confluency, HASMCs were treated with Rhodamine-labeled neutral PLPs and R8-PLP groups at 100 uM total lipid in DMEM. After 30 min and 24 h liposome exposure, cells were washed three times in PBS and qualitatively imaged by fluorescent microscopy. Images were acquired with a Texas Red fluorescent filter at 400× under 400 mse exposure across all groups (BX51 Olympus microscope, Olympus Q-color camera, Olympus Corporation, Shinjuku, Tokyo, Japan).

### 2.6. Cytotoxicity Analysis

At ~60% confluence, HASMCs were treated with neutral PLPs and R8-PLP groups at a concentration range of 25–100 uM total lipid in DMEM for 24 h. LIVE/DEAD^®^ Viability/Cytotoxicity Kit (Thermofisher Scientific) was used to determine relative cell toxicity, according to the manufacturer’s instructions. Briefly, DMEM was removed, cells were washed twice in PBS and co-stained with calcein-AM + ethidium homodimer for 15 min at 37 °C. Stained cells were visualized via fluorescein isothiocyanate (FITC; live) and Texas Red (dead) fluorescent filters. Images were acquired on both filters at 100×, using the same image acquisition system described above and cells were counted using ImagePro software (Media Cybernetics, Inc., Rockville, MD, USA). Cellular toxicity was calculated as [dead cell count/(dead cell count +live cell count)] in each independent sample and the mean normalized to the baseline cell death in non-treated controls of each experimental replicate.

### 2.7. Statistical Analysis

All data are reported as mean ± SEM. Statistical analyses were performed using Student’s *t*-test or one-way ANOVA and a post-hoc Student-Newman-Keuls test using SPSS 25 software (Systat Software, Inc., San Jose, CA, USA). Probability (*p*) values ≤0.05 were considered to be significant.

## 3. Results

### 3.1. Incorporation of R8-PEG via Pre-Insertion, Post-Insertion and Post-Conjugation Resulted in Significant siRNA Leakage and Reduced Total siRNA Retention when Compared to Unmodified PLP Controls

Modification of neutral PLPs with 5 mol% R8-PEG via post-insertion at 4 and 37 °C and via post-conjugation at 4 and 37 °C resulted in 5.11 ± 1.64%, 1.63 ± 0.89, 6.66 ± 2.80% and 4.98 ± 1.78% final siRNA EE%, respectively (Figure 2). Pre-insertion of 5 mol% R8-PEG resulted in 16.61±7.06% total siRNA retention, significantly higher than all other R8-PEG groups (* *p* < 0.05 vs. other R8-PEG groups; n = 5–6; Figure 2), but still significantly lower than unmodified PLP controls. Additionally, incorporation of R8-PEG by both pre-insertion and post-conjugation resulted in increased nanoparticle size and/or aggregation and increased PDI (Table 3).

### 3.2. Incorporation of STR-R8 via Pre-Insertion Resulted in Significantly Enhanced siRNA Encapsulation and Retention above All Other R8-Modified Plps and PLP Controls

Pre-insertion of 5 mol% STR-R8 into base PLP resulted in 55.86 ± 6.12% total siRNA encapsulation, significantly higher than all other STR-R8 modification techniques, all R8-PEG modification groups and unmodified PLP controls (**P < 0.05 vs. all other groups; n = 3–5; Figure 2). Additionally, pre-insertion of STR-R8 resulted in a more favorable nanoparticle characterization profile of size, PDI and charge compared to other groups (Table 3).

### 3.3. Ca^2+^-Mediated Etoh Injection of R8-Plps Resulted in Homogenous Liposome Samples between 50–60 nm with Increased siRNA Retention and EE%

We have previously reported that STR-R8 increases EE% in a dose dependent manner, with 10 mol% resulting in the highest of all tested modification levels [22]. Here, 10 mol% STR-R8 was pre-inserted into our R8-PLP assembly, both with and without the presence of Ca^2+^ at injection. STR-R8 incorporation without calcium results in EE% of 35.24 ± 2.18%, while the addition of 10 mM Ca^2+^ significantly increased EE% to 66.28 ± 9.14% (** *p* < 0.05 vs. 0 mM Ca^2+^; n = 4; Figure 3A). The addition of 20–50 mM Ca^2+^ also increased EE% above 0 nM, but not to a degree greater than 10 mM. Therefore, R8-PLP siRNA EE% is synergistically enhanced by STR-R8 incorporation and assembly in the presence of calcium. Additionally, the combinatorial addition of Ca^2+^ and STR-R8 had no effect on nanoparticle size, with all liposome groups at 50–60 nm in diameter (Figure 3B). R8-PLP samples had a slightly elevated size distribution compared to PLP controls, but PDI values still <0.25 (Figure 3C).

### 3.4. Increasing Lipid:siRNA (Wt-To-Wt) Ratios Enhanced R8-PLP siRNA EE%

Using the empirically defined assembly parameters for the incorporation of 10 mol% STR-R8 via pre-insertion in the presence of 10 mM Ca^2+^ at injection, the effect of lipid:siRNA wt-to-wt ratio was tested from 5:1–100:1. Ratios of 5, 10, 20, 50 and 100:1 achieved EE% of 59.31 ± 4.26%, 77.22 ± 3.42%, 80.64 ± 2.36%, 90.62 ± 4.76% and *** 97.06 ± 5.34%, respectively (*** *p* < 0.05 vs. all groups ≤ 20:1; n = 6; Figure 4A). Gel exclusion assays ran on unpurified liposome samples showed siRNA bands of decreasing intensity, representing decreasing amounts of un-encapsulated, un-complexed siRNA with increased lipid:siRNA (Figure 4B). Notably, the siRNA band representing the 100:1 sample was almost fully excluded from the gel, indicating ~100% complexation of siRNA with the R8-PLP nanoparticles (Figure 4B).

### 3.5. Partial Heparin Displacement Indicated siRNA Complexation to R8-PLP Surface in Addition to R8-PLP Entrapment within

Heparin displacement assays using 100:1 R8-PLP samples, assembled as empirically defined, demonstrated that treatment with 100 ug/mL heparin for 30 min at RT resulted in maximal siRNA displacement from the liposome surface (Figure 5A). When lipid:siRNA samples (5:1–100:1) were treated accordingly for full displacement, faint siRNA bands were present in all samples with increasing band intensity directly related to increasing lipid:siRNA. This indicated an increasing proportion of total siRNA EE% should be attributed to outer associated siRNA on the R8-PLP surface (Figure 5B panel 1). When equal fraction samples were treated with heparin+Triton X-100, for denaturation and displacement, strong siRNA bands further increased in intensity with increasing lipid:siRNA (Figure 5B panel 2), confirming increasing total siRNA EE% with increasing lipid:siRNA. Additionally, these bands were significantly more intense than equivalent lipid:siRNA samples treated with heparin only, indicating the proportion of outer associated siRNA is considerably less compared to total EE%. When lipid:siRNA samples were treated with heparin and re-dialyzed overnight at 4 °C to remove outer siRNA, then retreated with heparin+Triton X-100, equal fraction bands representing only internal encapsulated siRNA also increased in direct relation to lipid:siRNA (Figure 5B panel 4). Although band intensity of gels representing inner (panel 4) vs. outer (panel 1) siRNA cannot be objectively compared due to the presence of Triton, as Triton’s UV absorption profile spans that of protein and nucleic acid detection from ~280–400 nm, it is reasonable to suggest that a considerable proportion of total siRNA encapsulate was internally protected and retained (Figure 5B panel 4).

### 3.6. R8-PLP Nanoparticles Sufficiently Protected Encapsulated and Complexed siRNA against RNase A Degradation.

RNase stability assays against free siRNA controls demonstrated that treatment with 0.5 ug/mL RNase A enzyme for 30 min at 37 °C was sufficient for complete degradation of free siRNA at concentrations used within nanoparticles assays (Figure 6A). When R8-PLPs, assembled as empirically defined at 100:1, were treated with 0.5 ug/mL RNase prior to denaturation, there was no demonstrable siRNA degradation, as evidenced by equivalent siRNA bands in denatured R8-PLPs without RNase treatment (Figure 6B). Likewise, free siRNA control samples simultaneously exposed to RNase were completely degraded, indicating complete RNase protection of encapsulated and/or complexed siRNA by the R8-PLP nanoparticle.

### 3.7. Slower Injection Rates during R8-PLP Assembly Resulted in Increased Sample Homogeneity but Showed No Correlation to EE%

Using the empirically defined parameters for R8-PLP assembly, varying the speed of injection at assembly had no significant effect on EE% (n = 4; Figure 7A). Slower injection rates also resulted in an increased average diameter of the nanoparticles, but all remained ≤65 nm (n = 3; Figure 7B). Importantly, slower injection rates resulted in a more homogeneous nanoparticle population, with PDI < 0.2 at 0.1 and 0.2 mL/min (n = 3; Figure 7C).

### 3.8. R8-PLP Nanoparticle Assembly, at The Specifications Empirically Defined Herein, Is an Efficient Method to Produce Uniform, Unilamellar R8-Modified Liposomes

Imaging of R8-PLP lipsosomes via STEM, using a negative-stain method, revealed spherical morphology and single lamellarity (Figure 8). Images also confirm R8-PLP size reported by DLS data in Table 3 (italicized).

### 3.9. R8-PLP Nanoparticles Demonstrated Enhanced Cell Association In Vitro

Rho-PE-labeled R8-PLPs and PLP controls were assembled using the empirically defined parameters defined herein at a lipid:siRNA ratio of 20:1. Qualitative fluorescent images of HASMCs treated with these lipsome groups for 30 min and 24 h exhibited enhanced cellular association of R8-PLP nanoparticles compared to PLP controls (Figure 9).

### 3.10. R8-PLP Nanoparticles Demonstrated Negligible Cytotoxoic Effects across Lipid Conentration Ranges Appropriate for siRNA Delivery

R8-PLP and PLP controls were assembled identical to those utilized in the above cellular association assays, but without the Rhod-PE fluorescent label. When HASMCs were exposed to R8-PLPs and PLP controls for 24 h, then subjected to Live/Dead (green/red) dual cytotoxicity staining, qualitative images via fluorescence microscopy revealed negligible cytotoxic effects in any group (Figure 10A). Quantitative analyses indicated exposure of HASMCs to PLPs resulted in <1% cell toxicity above basal cell death across all concentrations after 24 h in culture. Exposure to R8-PLPs demonstrated more cytotoxic effects over PLP controls, but remained at <6% cell toxicity even at the upper concentration limit. (Figure 10A,B).

## 4. Discussion

Therapeutic strategies utilizing siRNA have been validated in a broad spectrum of disease pathologies in vitro [23,24]. Unfortunately, the clinical success of siRNA-mediated gene therapeutics is hampered by inadequate delivery technology [25]. Liposomes have emerged as the most widely tested nanocarriers of siRNA due to biocompatibility, flexibility of assembly and multifunctional potential [2,26]. However, effective liposome-mediated transfection typically requires cationic lipids to improve nucleic acid EE% and cell associative properties [27]. Unfortunately, CLP-induced cytotoxicity and immunogenicity has limited translational efficacy of liposomal gene therapy in vivo [28]. Therefore, non-cationic PLPs require optimization for enhanced EE% and cell interactive properties in order to achieve the translational success that remains elusive in liposome-mediated gene therapy studies. Here, we aimed to establish a noncationic PLP formulation with natural lipid constituents and surface-conjugated CPP as an improved transfection agent compared to traditional CLP formulations that often fail in downstream gene therapy applications. The development of a simple, scalable method for assembling noncationic CPP-modified PLPs capable of efficient siRNA loading and enhanced cell association could provide an ideal drug delivery system for gene therapy.

R8 is a well-established CPP that can be incorporated into PLP bilayers using R8-amphiphiles to expose the active peptide on the liposome surface. Examples of R8-amphiphiles include R8 conjugated to DSPE-PEG (R8-PEG), R8 anchored to stearic fatty acid (STR-R8) and others [29,30]. Common liposome modification techniques that can be used for R8-amphiphile incorporation are (1) inclusion of R8-amphiphile during initial liposome assembly (pre-insertion), (2) insertion of R8 amphiphiles into the bilayers of pre-formed liposomes (post-insertion) and (3) reacting the R8 peptide with functional groups exposed on the surface of pre-formed liposomes utilizing click chemistry techniques (post-conjugation) [31,32]. However, process parameters required to assemble CPP-modified noncationic PLPs with efficient siRNA loading and functionally enhanced delivery are previously unknown.

Here, we first examined the modification of non-cationic PLPs with R8 by incorporating R8-PEG into the liposomal formulation via pre-insertion, post-insertion and post-conjugation techniques. One liposomal attribute that directly affects drug leakage during ligand modification is membrane fluidity [33]. Therefore, in order to minimize membrane fluidity of pre-formed liposomes during R8 modifications, we tested post-insertion and post-conjugation efficiency at 4 °C in addition to previously established reaction conditions of 37 °C [34,35]. With 4 °C still being well above phase-transition of liposomes comprised of unsaturated lipids, it was expected that both 4 °C and 37 °C would provide the thermal energy required for adequate insertion. This allowed for the assessment siRNA leakage and retention upon R8-PEG incorporation based on the effects of temperature on membrane fluidity. As expected, when performing post-insertion at 4 °C for increased membrane stability, a slightly higher retention of siRNA was achieved when compared to post-insertion at 37 °C, but final EE% was still only ~5% (Figure 2). Post-conjugation is often avoided when dealing with CPPs, due to their membrane perturbing effects and potential for siRNA leakage upon incorporation. Here, we confirmed that using R8 with unnatural lysine(azide) residues to decorate the surface of pre-formed PLPs by post-conjugation resulted in significant siRNA leakage irrespective of temperature. In fact, the final siRNA encapsulation for post-insertion and post-conjugation in all conditions was no higher that ~7%, limiting the overall feasibility of assembling R8-modified PLPs via post-insertion or post-conjugation of R8-PEG. Ultimately, pre-insertion of R8-PEG resulted in slightly enhanced encapsulation above all other R8-PEG groups at ~17% (Figure 2), but the PDI of these liposomes was significantly higher even after extrusion (Table 3). Peptides have more degrees of freedom when conjugated to PEG residues compared to direct lipid conjugation, largely due to the many degrees of freedom that the polyethylene glycol polymer exhibits under physiological conditions [36]. It is possible that this highly dynamic scenario of the R8 peptide interferes with assembly during pre-insertion. Therefore, while the encapsulation of siRNA is modestly increased during liposome assembly using R8-PEG pre-insertion, compared to post-insertion or post-conjugation techniques, the efficient assembly of siRNA nanocarriers is still compromised due to low overall EE%.

Next, we examined the modification of non-cationic PLPs with STR-R8. With R8 anchored to the stearic fatty acid, the entrapment of liposomal siRNA was highly efficient. Contrary to pre-insertion using R8-PEG, the pre-insertion of STR-R8 had little effect on size dispersity (PDI < 0.25) and still resulted in liposomes below 50 nm (Table 3). Furthermore, pre-insertion of STR-R8 enhanced the overall encapsulation of siRNA to ~56%, the highest EE% achieved when comparing all R8-amphiphile incorporation techniques (Figure 2). To our knowledge, there is no established one-step method to encapsulate siRNA above ~50% efficiency in noncationic PLPs. The method demonstrated here is a one-step assembly whereby all lipid constituents, including ligand-conjugated lipids, are solubilized together in organic solvent prior to injection into aqueous buffer containing siRNA encapsulate. This one-step assembly method is inherently simpler than post-insertion or post-conjugation techniques, which require multiple assembly and purification steps. Harashima et al. previously demonstrated the unique ability of STR-R8 to condense nucleic acid through electrostatic complexation [37]. We propose that the enhanced EE% seen in R8-PLPs assembled via pre-inserted STR-R8 is a result of the condensation of siRNA during liposome assembly. A portion of the STR-R8 is integrated into the PLP bilayer to help enhance delivery, but much of it may also promote condensation and enhanced encapsulation of siRNA upon incorporation into PLPs. In this way, the STR-R8 moiety provides a bifunctional element to R8-PLP assembly and application.

To further elucidate the process parameters for R8-PLP formulation and assembly for optimal encapsulation of siRNA, we examined the synergistic contribution of Ca^2+^-mediated nucleic acid condensation. Similar to previously reported results, EtOH injection of non-modified PLPs in the presence of Ca^2+^ enhanced the EE% of siRNA with as little as 10 mM Ca^2+^ compared to PLPs with no Ca^2+^ When assembling R8-PLPs with pre-insertion of STR-R8 in the presence of Ca^2+^ there was a synergistic enhancement of siRNA EE% with ≥10 mM, whereby encapsulation approached ~70% at all concentrations and no deleterious effect was seen in nanoparticle size or homogeneity (Figure 3A–C). Moving forward, this one-step assembly technique for R8-PLPs, using STR-R8 modification via pre-insertion and 10 mM Ca^2+^ at injection, was utilized to assemble all liposome groups in order to define additional assembly parameters for optimized siRNA retention and enhanced cell association. A schematic representing this empirically defined and modified EtOH injection technique is presented in the graphical abstract.

For further optimization of assembly parameters, next siRNA loading and carrying capacity was examined by varying lipid:siRNA wt-to-wt ratios at injection. We demonstrated increased siRNA loading with increased lipid:siRNA, where 50:1 and 100:1 lipid:siRNA samples achieved >90% and >97% EE%, respectively (Figure 4A,B). In order to differentiate outer associated vs. inner entrapped siRNA proportions constituting this final EE% value, a series of gel electrophoresis assays were utilized. When all lipid:siRNA samples (5:1–100:1) were treated with heparin only for outer siRNA disassociation, faint bands indicated an increasing proportion of outer associated siRNA with increasing lipid:siRNA conditions (Figure 5B.1). However, when equivalent fractions of each lipid:siRNA sample were treated with heparin+Triton to release both outer associated and inner entrapped siRNA, strong siRNA bands further increased in intensity, confirming increasing total siRNA EE% with increasing lipid:siRNA (Figure 5B.2). Subsequently, when additional equivalent fractions were heparin treated and re-dialyzed overnight to clear outer associated siRNA from the samples, these re-dialyzed lipid:siRNA samples were treated with heparin+Triton to release total remaining siRNA encapsulate from the inner core. Although band intensity of gels representing inner vs. outer siRNA cannot be objectively compared due to the presence of Triton, it is reasonable to suggest that a considerable proportion of total siRNA encapsulate was internally protected and retained (Figure 5B.4).

Given that a fraction of the nearly 100% encapsulation efficiency demonstrated here in our 100:1 sample could be partially attributed to outer associated nucleic acid, one could postulate that the delivery efficacy of our empirically derived R8-PLP platform described herein might be overestimated for in vitro and/or in vivo translation due to nucleic acid instability in the presence of serum. Therefore, to determine R8-PLP stability and siRNA protection, R8-PLPs with the highest EE% (assembled as described at 100:1) were subjected to RNase A enzymatic degradation prior to gel electrophoresis. As shown, R8-PLPs exposed to RNase degradation exhibited siRNA bands of equal intensity to untreated R8-PLP controls, while free siRNA was completely degraded (Figure 6B). This qualitative assay confirmed complete protection of siRNA when encapsulated and/or entrapped within our R8-PLP nanocarrier using the assembly method described herein.

Finally, to test the dual-functionality of the STR-R8 in increasing siRNA EE% while simultaneously enhancing the cell associative properties noncationic PLP assemblies, HASMCs were treated with R8-PLPs and PLP controls. Here, Rhodamine-labeled lipid (0.5 mol%) was added to PLP and R8-PLP assemblies for liposome tracking. As revealed through fluorescent microscopy, R8-PLPs significantly enhanced HASMC association at both 30 min and 24 h post treatment in vitro compared to unmodified PLP controls (Figure 9). Likewise, in vitro cytotoxicity analyses revealed that lipid nanoparticle exposure across 25–100 uM total lipid concentrations were tolerable for application, without notable deleterious toxic thresholds either in R8-PLPs or PLP controls. Importantly, in considering the demonstrated siRNA encapsulation efficiency of R8-PLPs reported herein, the tested lipid concentration range would correspond to approximately 75–300 nM siRNA encapsulate applied. This falls well within reasonable molecular biology protocols for pharmaceutical and chemical siRNA transfection using commercially available reagents, further supporting the feasibility of this R8-PLP nanoparticle for effective gene therapy delivery.

Of note, a major focus of our research program is aimed at developing lipid nanoparticles for targeted vascular gene therapeutics. Although this present study is not focused specifically on vascular application, the HASMCs used herein are a primary cell type of experimentation in our laboratory. In the field of molecular biology, this cell type is considered a generally hard-to-transfect primary cell line. As such, the successful application of R8-PLP nanoparticles in HASMCs presented here is promising for future empirically derived studies in a multitude of different cell and tissue types. Consequently, we believe the cell-type selection and assays described here are a strength of the current study and that this preliminary in vitro validation supports the need for further studies of functionality and delivery efficacy in vitro and in vivo. As such, our future assays will focus on in vitro gene targeting efficacy and ex vivo and in vivo delivery efficacy, analyses that will inherently determine the applicability of R8-PLPs in less controlled experimental environments indicative of the variable milieu encountered in translational implementation. Additionally, other future pre-clinical studies are planned for in vivo biodistribution analyses and pharmacokinetic profiling of the R8-PLP nanocarrier platform.

Our ultimate goal is to elucidate the most efficient method for producing CPP-modified PLPs that have the potential for downstream gene therapy applications. To this end our methodology must have the potential for scalable production and the resultant delivery platform must demonstrate functionality in a dynamic biological environment. While our laboratory manually derived and validated many of these specifications, the development of instrumentation to control these and other parameters is a source of ongoing collaborative efforts among our group and others. Consequently, the data presented here regarding controlled injection rates and the absence of a significant effect on EE% may point to an assembly method that could be highly scalable for industrial applications.

## 5. Conclusions

In conclusion, our research has demonstrated that the addition of STR-R8 into noncatonic PLPs utilizing the pre-insertion technique is an efficient method to produce unilamellar R8-modified liposomes capable of enhanced cell association and efficient siRNA encapsulation and retention. This development could provide a foundation for future clinical applications of liposomal gene therapy with greater translational success.

## Figures and Tables

**Figure 1 nanomaterials-11-00825-f001:**
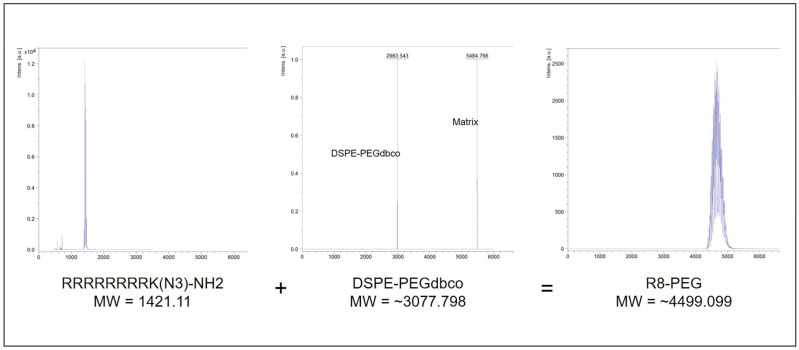
Mass spectroscopy confirmed R8 peptide conjugation to DSPE-PEGdbco via azide-alkyne cycloaddition reaction (i.e., click chemistry).

**Figure 2 nanomaterials-11-00825-f002:**
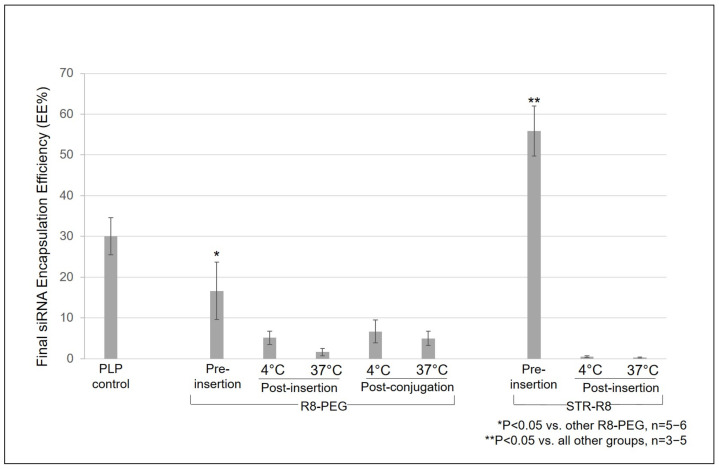
Incorporation of STR-R8 via pre-insertion resulted in the highest siRNA EE% above all.

**Figure 3 nanomaterials-11-00825-f003:**
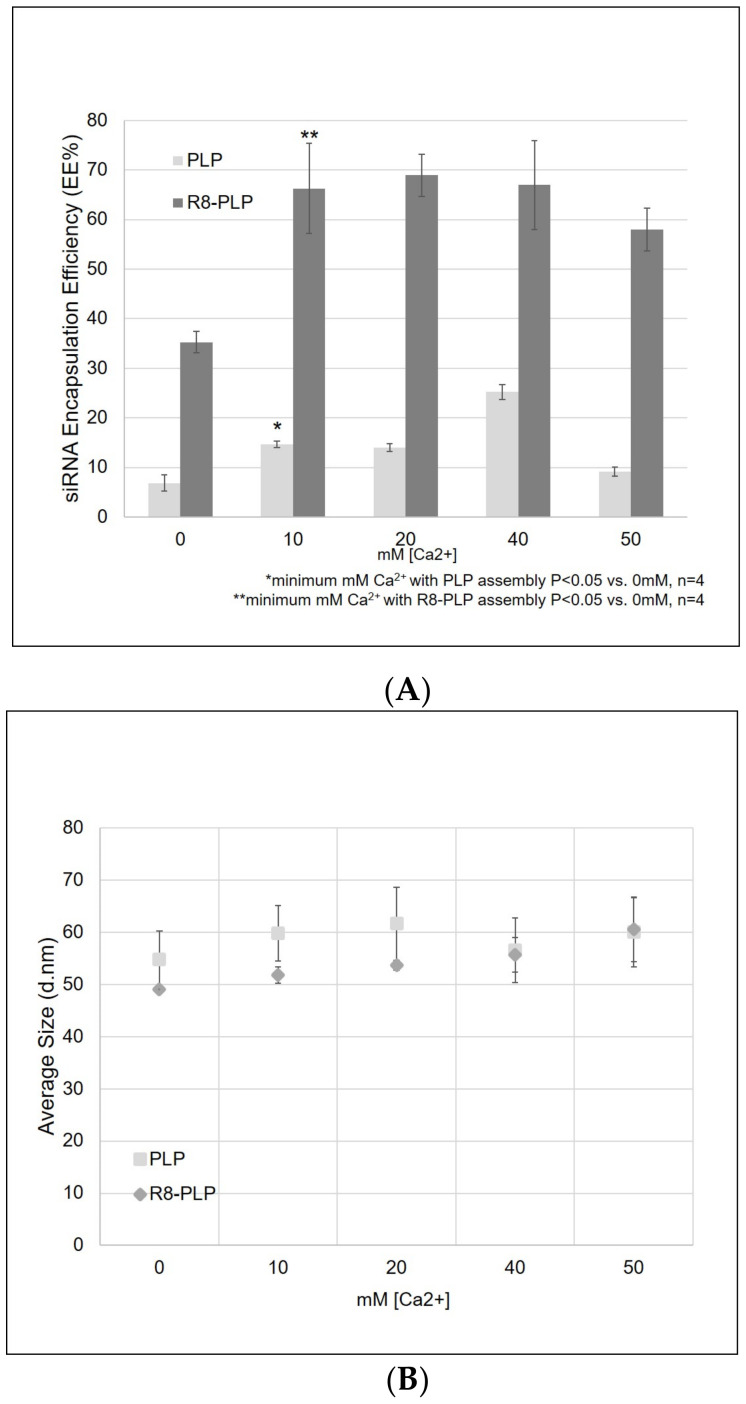
Ca^2+^-mediated EtOH injection of R8-PLPs resulted in (**A**) increased siRNA retention and EE% in (**B**) 50–60 nm liposome samples with a (**C**) homogeneous size distribution.

**Figure 4 nanomaterials-11-00825-f004:**
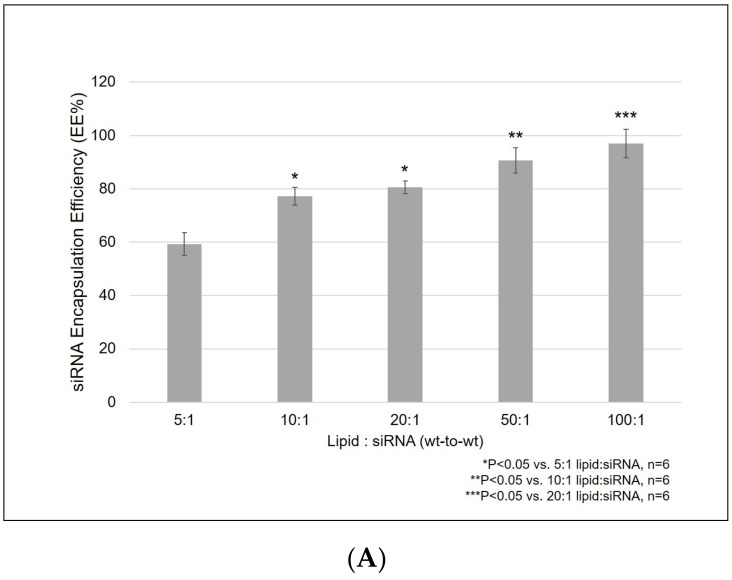
(**A**) Increasing lipid:siRNA (wt-to-wt) ratios enhanced R8-PLP siRNA EE% while (**B**) gel exclusion assays ran on unpurified liposome samples served as physical confirmation of increasing siRNA EE%.

**Figure 5 nanomaterials-11-00825-f005:**
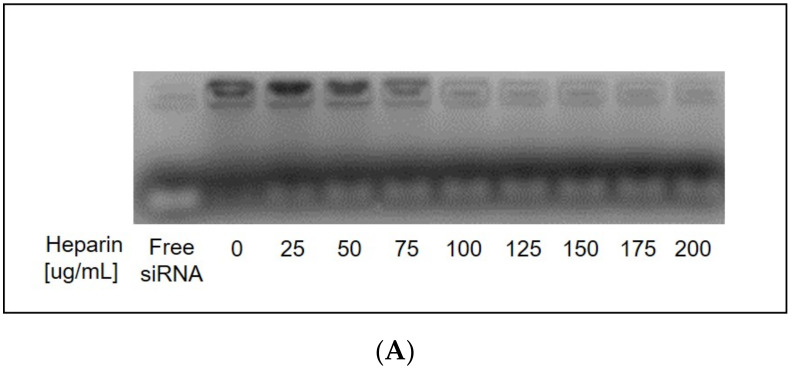
(**A**) Assay development indicated treatment with 100 ug/mL heparin was the minimum concentration required for effective removal of all outer-associated siRNA. (**B**) Heparin displacement assays indicated siRNA complexation to R8-PLP surface (panel 1), total EE% (panel 2), and siRNA entrapment within the R8-PLP core (panels 3 and 4), in a lipid:siRNA dependent manner.

**Figure 6 nanomaterials-11-00825-f006:**
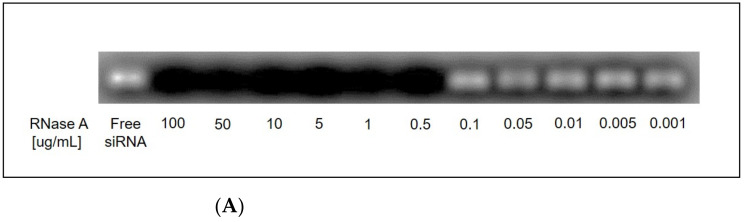
**(A**) Assay development indicated treatment with 0.5 ug/mL RNase A enzyme was the minimum concentration sufficient for complete siRNA degradation. (**B**) RNase stability assays indicated that R8-PLP nanoparticles sufficiently protected encapsulated and complexed siRNA when exposed to nuclease digestion.

**Figure 7 nanomaterials-11-00825-f007:**
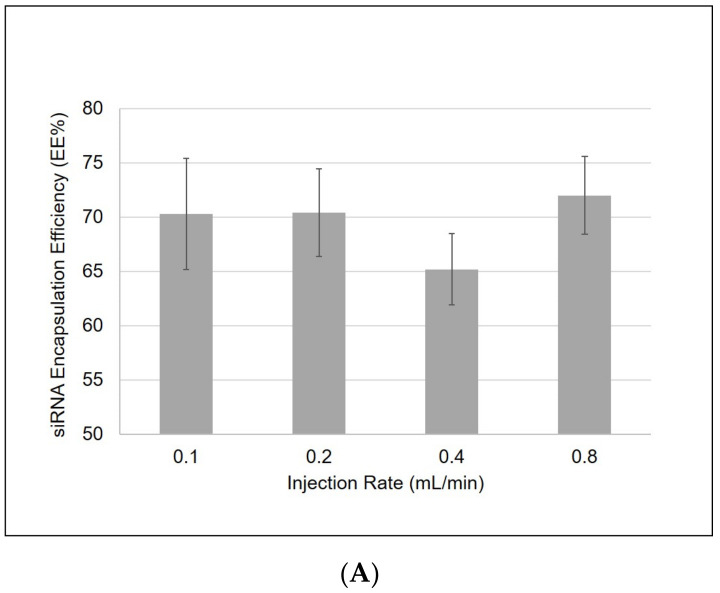
Slower injection rates during R8-PLP assembly (**A**) had no significant effect in siRNA EE%, but resulted in (**B**) increased mean diameter and (**C**) lower PDIs.

**Figure 8 nanomaterials-11-00825-f008:**
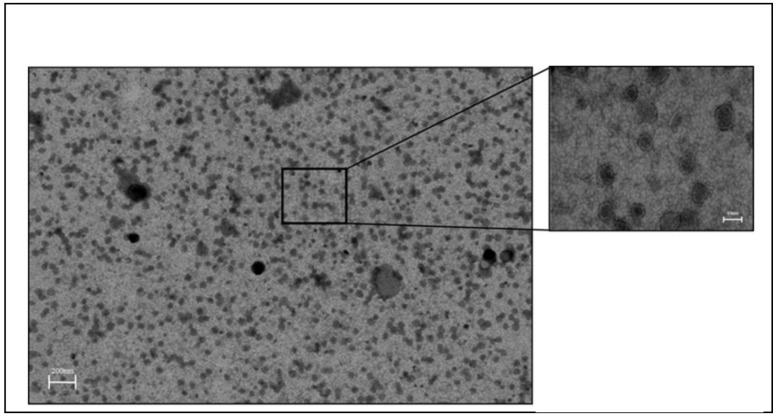
STEM images of R8-PLPs.

**Figure 9 nanomaterials-11-00825-f009:**
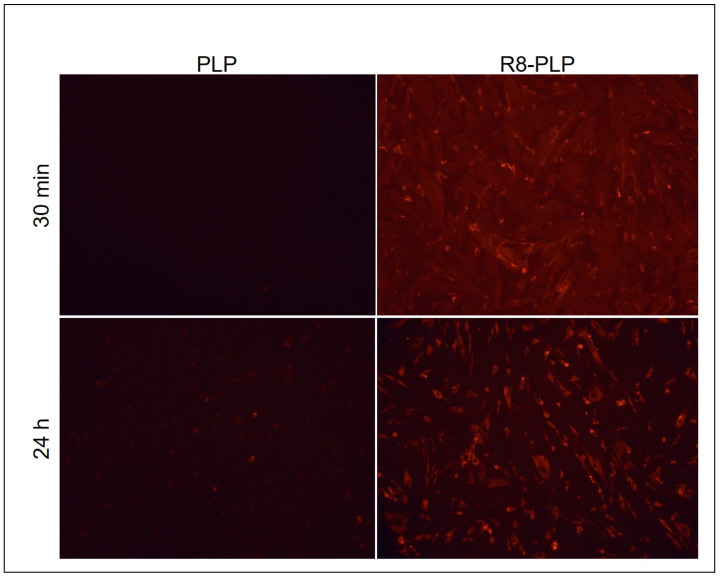
R8-PLP nanoparticles demonstrated enhanced cell association in vitro.

**Figure 10 nanomaterials-11-00825-f010:**
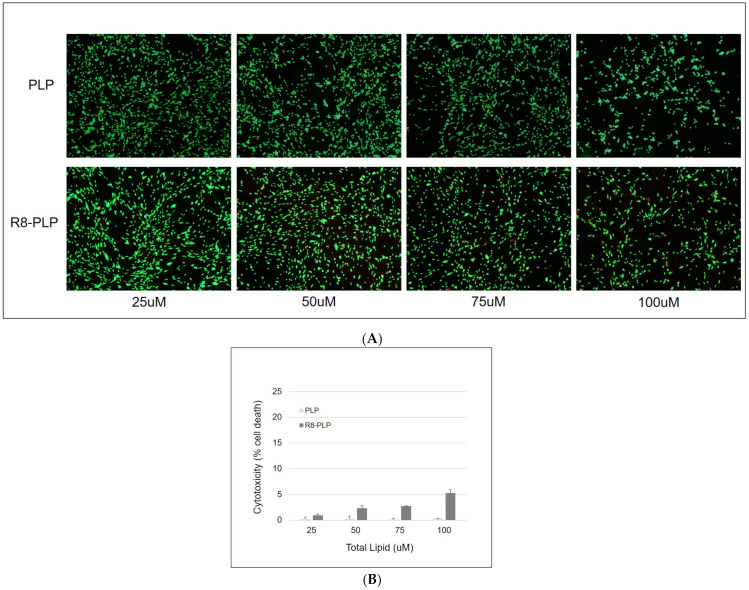
R8-PLP nanoparticles demonstrated negligible cytotoxic effects in vitro under (**A**) qualitative and (**B**) quantitative analyses.

**Table 1 nanomaterials-11-00825-t001:** Quality by design approach.

Critical Quality Attribute (CQA)	Units Measured	Desired Range
Encapsulation Efficiency (EE%)	% total siRNA	>50%
Zeta Potential (ζ)	millivolts (mV)	<10 mV
Average Size	diameter (nm)	<100 nm
Polydispersity Index (PDI)	N/A	<0.2

**Table 2 nanomaterials-11-00825-t002:** PLP and R8-PLP Lipid Formulation Constituents.

Lipid Constituent	Acronym
1,2-dioleoyl-sn-glycero-3-phosphocholine	DOPC
Cholesterol	chol
1,2-distearoyl-sn-glycero-3-phosphoethanolamine-N-[amino(polyethylene glycol)-2000]	DSPE-PEG
1,2-distearoyl-sn-glycero-3-phosphoethanolamine-N-[dibenzocyclooctyl(polyethylene glycol)-2000]	DSPE-PEGdbco
1,2-dioleoyl-sn-glycero-3-phosphoethanolamine-N-(lissamine rhodamine B sulfonyl)	Rho-PE
octaarginine (RRRRRRRRK(N3)-NH2)	R8

**Table 3 nanomaterials-11-00825-t003:** Liposome characterization related to R8 modification and assembly conditions.

	Modification Conditions		Pre-Modification			Post-Modification		
	R8-Lipid Conjugate	Modification Type	Size (d.nm)	PDI	ζPotential (mV)	Size (d.nm)	PDI	ζPotential (mV)
**PLP**	**N/A**	N/A	55.84 ± 5.00	0.25 ± 0.04	9.74 ± 1.02	55.84 ± 5.00	0.25±0.04	9.74 ± 1.02
**R8-PLP**	**R8-PEG**	Pre-insertion	N/A	N/A	N/A	820.8 ± 375.7	0.52 ± 0.13	9.93 ± 1.44
	**R8-PEG**	Post-insertion (4 °C)	58.32 ± 1.75	0.18 ± 0.012	10.21 ± 1.16	68.01 ± 5.89	0.217	10.72 ± 2.15
	**R8-PEG**	Post-insertion (37 °C)	61.75 ± 3.17	0.22 ± 0.02	10.70 ± 1.48	72.55 ± 5.45	0.180	11.92 ± 1.45
	**R8-PEG**	Post-conjugation (4 °C)	59.70 ± 6.14	0.2 1 ± 0.05	10.44 ± 1.44	176.78 ± 47.89	0.313	10.64 ± 1.27
	**R8-PEG**	Post-conjugation(37 °C)	55.98 ± 5.47	0.21 ± 0.04	10.58 ± 1.49	172.4 ± 52.1	0.354	9.11 ± 0.94
	**STR-R8**	Pre-insertion	N/A	N/A	N/A	**49.65 ± 1.77**	**0.22 ± 0.01**	**7.40 ± 1.90**
	**STR-R8**	Post-insertion (4 °C)	51.14 ± 0.97	0.16 ± 0.01	1.54 ± 6.87	53.74 ± 2.37	0.21 ± 0.02	13.70 ± 2.63
	**STR-R8**	Post-insertion (37 °C)	49.00 ± 1.75	0.21 ± 0.01	8.54 ± 2.35	48.83 ± 2.00	0.21 ± 0.01	12.10 ± 0.55

## Data Availability

Data are contained within the article.

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
