# Peer review of "Advances in the Formulation and Assembly of Non-Cationic Lipid Nanoparticles for the Medical Application of Gene Therapeutics"

_nanomaterials, 2021, doi:10.3390/nano11030825_

Round 1
Reviewer 1 Report
The submitted manuscript describes the comparison of various protocols to encapsulate siRNA in liposomes using the cell-penetrating peptide octaarginine R8. The authors differentiated between encapsulated siRNA and complexation to the particle surface, which is very important because encapsulation is very important for preservation of siRNA integrity. After physicochemical characterization the fluorescence-loaded liposomes is localized in cultures of arterial smooth muscle cells. The data will support the conclusion by the authors that the presented technique can be used to efficiently encapsulate siRNAs if some additional information is added.
Comments:
- Were liposomes characterized in DMEM, in which they were applied, or only in distilled water? Please list both.
- Why were smooth muscle cells used for the testing and why were they only assessed in growth arrested state? In the end, uptake into the cells is intended because the siRNA has to act in the cytoplasm.
- Even though liposomes in general are not highly cytotoxic, cytotoxicity has to tested to identify the concentration range, in which they could be used. Therefore, viability of the cells should be presented.
Reviewer 2 Report
Very interesting article on formulation.
Three small editorial notes:
-digit formatting that should be in superscript or subscript is missing, e.g. CaCl2, CHCl3, N2, Ca2+
-description of the drawing "Figure 1, 2 ..." is below the drawing. There is no need to add this information to the drawing itself
-Table 3 - in line concerning PLP, data concerning pre-modification and post-modification are the same, besides "size". Should not they all be the same?
Round 2
Reviewer 1 Report
Thank you for the revision, I have no further comments.